# Dissecting the shared genetic architecture of bipolar disorder, major depressive disorder, and attention-deficit hyperactivity disorder

Christopher Lawrence[1,2,3]◉*, Thomas Folkmann Hansen[3,4]◉

**1** Wake Forest Department of Biology, Winston-Salem, North Carolina, United States of America, **2** Information Systems and Wake Forest University. WFU High Performance Computing Facility, Winston-Salem, North Carolina, United States of America, **3** DIS Research, Copenhagen, Denmark, **4** Neurogenomic, Translational Research Centre, Copenhagen University Hospital, Glostrup, Denmark

◉ These authors contributed equally to this work.
* chrislaw133@gmail.com

## Abstract

Major depressive disorder (MDD), bipolar disorder (BPD), and attention-deficit hyperactivity disorder (ADHD) are prevalent, highly heritable psychiatric disorders with significant degrees of genetic overlap. Using open-sourced summary statistics from the Psychiatric Genomics Consortium and 1000 Genomes European reference panel, we fit a latent factor model (F1) capturing the shared genetic liability across MDD, BPD, and ADHD. Multivariate GWAS identified 350 linkage disequilibrium-independent loci, 105 of which have not been previously reported by the contributing univariate GWASs. Univariate, bivariate, and trivariate mixture models elucidated both shared and trait-specific polygenicity across the disorders. Gene-level analysis with Hi-C coupled MAGMA identified a total of 2936 novel dopaminergic associations across the constituent disorders that went undetected in univariate analyses. Among the top genes associated with F1, protein tyrosine phosphatase receptor type D emerged as a promising candidate. Cell typing and brain tissue enrichment for F1 further implicated the cerebellum and cholinergic neurons. These findings demonstrate how multivariate approaches can elucidate shared biological mechanisms, providing new etiological insights into individual disorders and implicating therapeutic targets for the treatment of psychiatric comorbidity.

## Introduction

The World Health Organization estimates that 5% of adults suffer from major depressive disorder globally [1]. The Diagnostic and Statistical Manual of Mental Disorders, 5th edition, characterizes major depressive disorder as having 5 or more of the following symptoms: depressed mood, loss of interest in pleasurable activities, weight loss or gain, sleep disturbances, psychomotor agitation or retardation, fatigue,

provided the original author and source are credited.

**Data availability statement:** Supplementary Information and S1–S16 Figs can be accessed at https://zenodo.org/records/18293051/files/ Supplementary%20Information.docx?down- load=1. S1–S34 Tables can be accessed at https://zenodo.org/records/18148979/files/ Supplementary_Tables.xlsx?download=1. All scripts used in this paper can be found at https://github.com/chrislaw133/PTPRD. The multivariate summary statistics can be down- loaded at https://zenodo.org/records/16741578/ files/F1.sumstats.txt.gz?download=1. All data analyzed in this study can be found in the published articles' data repositories (See references).

**Funding:** The author(s) received no specific funding for this work.

**Competing interests:** The authors have declared that no competing interests exist.

feelings of worthlessness or excessive guilt, decreased concentration, and thoughts of suicide or death [2]. The etiology of Major depressive disorder is understood to arise from the complex interplay of biological, genetic, environmental, and psycholog- ical factors [3]. Genetic factors are strongly implicated in the development of major depressive disorder, and has been thoroughly demonstrated in twin, adoption, and family studies. The latest twin studies estimate the heritability to be between 40 and 50%, and family studies show that first degree relatives have a twofold to threefold higher lifetime risk of developing major depressive disorder [4].

The global prevalence of ADHD in adults and children is estimated to be 5% (The diagnosis and management of ADHD [5]. According to the Diagnostic and Statistical Manual of Mental Disorders, 5th edition, ADHD is diagnosed based on the amount and severity of symptoms associated with inattention, hyperactivity, and impulsivity. Genetic factors are heavily implicated in the development of ADHD, with a formal heritability of about 80% [6]. The latest twin studies have reported heritability to be between 77–88% [7].

It is estimated that 2.4% of the global population is affected by bipolar disorder [8]. Bipolar disorder is characterized by chronic alternation between manic and major depressive episodes [9]. Mania is distinguished by periods of unusually high energy and markedly elevated moods or emotions that diverge significantly from one's nor- mal state. Genetic factors are highly implicated in BPD, with the latest twin studies estimating the heritability to be 79–93% [10].

Mental disorder comorbidity is an extremely common phenomenon, and lifetime occurrences of mental disorders are highly correlated with increased risk for the onset and development of subsequent mental disorder(s) [11]. ADHD especially is a potentiator of comorbidity and is associated with increased MDD and BPD hazard ratios [12]. MDD is notably heterogenic, and frequently encompasses characteristic symptoms of bipolar disorder such as mania [13]. There is extensive genetic overlap across MDD, BPD, and ADHD [14].

GWASs have identified a substantial amount of reproducible risk loci associated with BPD, ADHD, and MDD [15–17]. Concurrently, there is significant degree of genetic overlap across the three disorders, complicating accurate diagnosis and treatment [18,19]. In this study, we investigate the shared genetic architecture of MDD, BPD, and ADHD, with the aim to examine pleiotropic mechanisms that underlie the development of these psychiatric disorders.

## Methods

The univariate GWAS summary statistics were acquired from the Psychiatric Genom- ics Consortium [15–17]. The PGC provides a plethora of open-sourced summary statistics from GWAS, excluding 23andme cohorts (Table 1). The most recent BPD, ADHD, and MDD GWAS cohorts of European ancestry were used to carry out this study. Practicing strict quality control, we removed all SNPs below a minor allele frequency threshold of 0.01 during summary statistics harmonization. The 1000 Genomes Project Phase 3 panel was used for reference of European genetic varia- tion [20] (Fig 1).

**Table 1. GWAS summary statistics sample sizes, $\lambda_{GC}$, mean $\chi^2$, associations (Genome-wide significant variants, p<5e-8), and independent associations (Genome-wide significant variants that are LD independent, p<5e-8 and $r^2<0.1$) downloaded from the Psychiatric Genomics Consortium.**

| Trait | Cases | Controls | $\lambda_{GC}$ | Mean $\chi^2$ | Associations | Independent Associations |
|---|---|---|---|---|---|---|
| MDD | 412, 305 | 1, 588, 397 | 1.897 | 1.089 | 14,446 | 32 |
| BPD | 59, 287 | 781, 022 | 1.496 | 1.652 | 3,017 | 81 |
| ADHD | 38, 899 | 186, 843 | 1.375 | 1.043 | 1,428 | 325 |

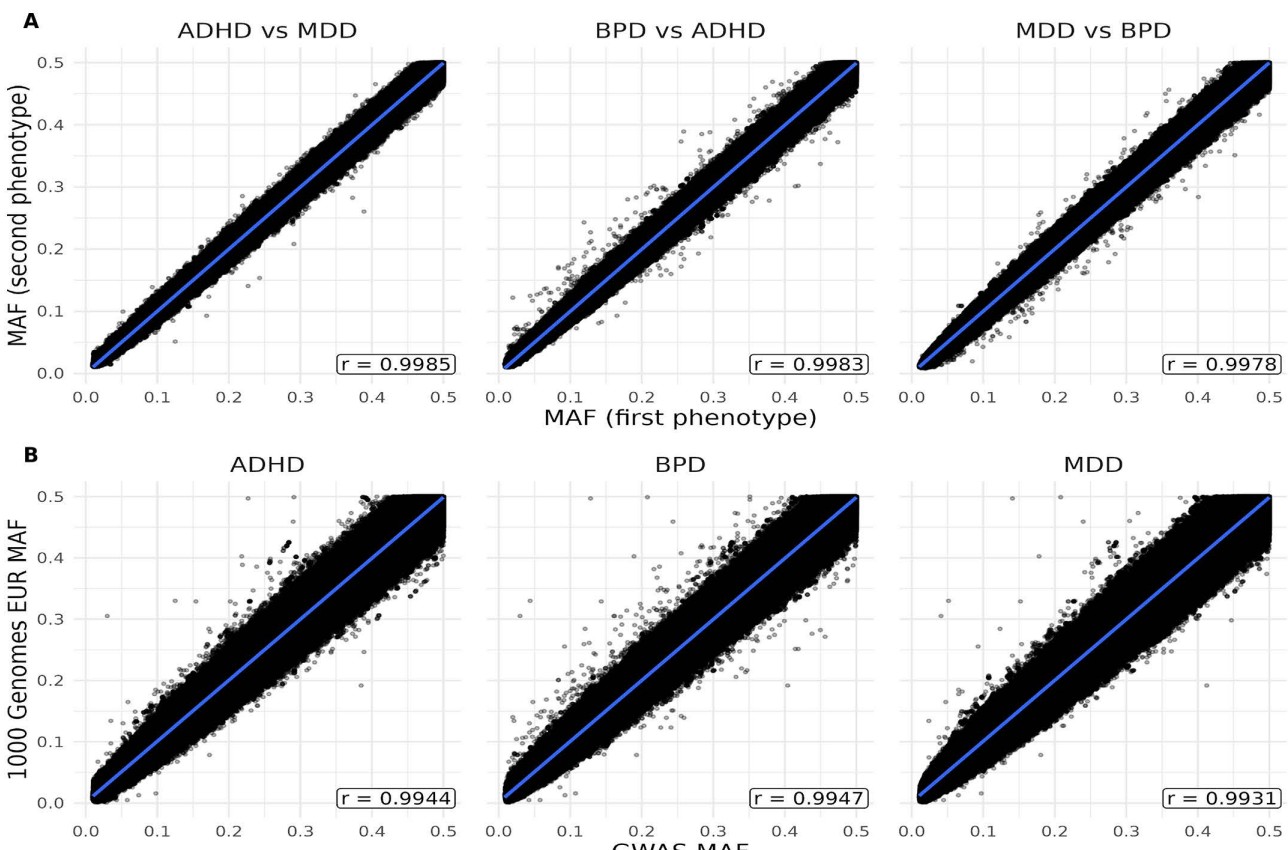

**Fig 1. Comparison of minor allele frequencies (MAF) across study cohorts and 1000G EUR reference panel. (A)** Plot of cohort vs. cohort minor allele frequencies (MAF), with Pearson correlations and 95% confidence interval from regression denoted by the blue line. **(B)** Plot of cohort vs. 1000 Genomes EUR minor allele frequencies (MAF), with Pearson correlations and 95% confidence interval from regression denoted by the blue line.

## MiXeR

The univariate, bivariate, and trivariate frameworks of MiXeR were used to estimate polygenicity across MDD, BPD, and ADHD. MiXeR is a causal mixture model that assumes a small fraction of variants have an effect on a trait(s), while the remaining variants have zero effect. These causal variant proportions are estimated as parameters by maximizing a log-likelihood function, with the objective of estimating parameters that best describe the observed z-score distributions of input GWAS summary statistics. The model of observed z-score distributions is weighted per-SNP by effective sample size, adjusting for sample size disproportionality. Additionally, the model's residual error term is specified as a zero-mean,

multivariate normal random vector with a freely estimated residual variance-covariance matrix that accounts for sample overlap and cryptic relatedness. The univariate, bivariate and trivariate models of MiXeR work under similar assumptions and allow us to conceptualize the genetic architecture of MDD, BPD, and ADHD by estimating both the number of causal variants needed to explain 90% of SNP heritability and causal variant discoverability [21,22].

### Genomic structural equation modeling

To gain functional insights into the shared genetic liability across MDD, ADHD, and BPD, the multivariate GWAS and TWAS extensions of genomic structural equation modeling (Genomic SEM) were used [23]. The frameworks utilize structural equation modeling to identify variants and tissue-specific genes that likely underlie the shared covariance across MDD, BPD, and ADHD. This process took place in two steps. The first step estimated total liability scale SNP heritability and genetic correlations between each trait using multivariable linkage disequilibrium score regression (LDSC). In the second step, a model is specified such that an unobserved variable F1, drives the shared genetic signal across ADHD, MDD, and BPD. In minimizing a weighted least squares objective, the chosen system of multiple regression and respective covariance associations across the disorders and F1 are estimated as model parameters that best align with the observed covariances from the univariate GWAS summary statistics. Genomic SEM populates a sampling covariance matrix using a jackknife resampling procedure with sampling errors of total liability scale SNP heritability and genetic covariances as diagonals and covariances of sampling errors on off diagonals. The sampling covariance matrix is utilized in a sandwich correction to robustly correct model parameter standard errors, accounting for unknown degrees of sample overlap. Additionally, each trait in the WLS objective is weighted by the inverse diagonals of the genetic sampling covariance matrix, accounting for sample size disproportionality. The trait-specific betas of each individual SNP or gene are then regressed onto the model to estimate an effect size on F1. Strict quality control procedures were applied to the F1 ~ SNP summary statistics for downstream analysis, including filtering out SNPs that pass $X^2$ and $Q_{SNP}$ genome-wide significance thresholds (p-value < 5e-8). PLINK was used to identify linkage disequilibrium independent SNPs ($r^2 < 0.1$) (Purcell et al. 2007).

### MAGMA/ H-MAGMA/ MAGMA cell typing

Secondary bioinformatics analysis was carried out on the F1 ~ SNP summary statistics. Hi-C coupled MAGMA (H-MAGMA) was utilized to aggregate F1 ~ SNP effect sizes to F1 ~ gene and gene-set effect sizes [24,25]. The midbrain, dopaminergic annotation was used to assess the shared genetic liability across MDD, BPD, and ADHD specifically for dopamine neurons. The top 30 protein coding gene hits were subject to further biological analysis. To characterize novel gene associations identified by F1, we used WebGestalt to carry out over-representation analyses [26].

To examine brain region and cell type associations for F1, MAGMA Cell Typing was used in tandem with the RNA expression dataset curated by Siletti et al. 2023 [27], which consists of over 3 million cell nuclei from 100 locations across the forebrain, midbrain, and hindbrain [28–30].

## Results

### Estimating polygenicity with MiXeR

The univariate, bivariate, and trivariate frameworks of MiXeR were used to model shared and trait specific causal variants across MDD, BPD, and ADHD, as well as estimate causal variant discoverability. ADHD's estimated number of causal variants to explain 90% of SNP heritability (nc@9p) was 7.81K (SD = 291) variants with a causal variant variance (Causal variant discoverability) of 3.00e-05 (SD = 1.03e-06). BPD's estimated nc@p9 was 7.96K (SD = 254) variants with a casual variant discoverability of 2.64e-05 (SD = 7.35e-07). MDD was the most polygenic and exhibited the smallest causal variant effect sizes across the three disorders, with an nc@9p of 12.18K (SD = 346) variants and a causal variant discoverability of 6.69e-06 (SD = 1.55e-07). The nc@9p for F1 was 11.03K (SD = 190) variants with a discoverability of 1.09e-05 (SD = 1.12e-07).

## Estimating polygenicity with MiXeR – Bivariate

There is an extensive amount of bivariate genetic overlap across the disorders (Fig 2). MDD exhibited strong genetic correlation with ADHD and BPD. However, despite BPD and ADHD having over 80% nc@9p overlap, a modest genetic correlation of 0.28 suggests that a significant proportion of shared causal variants are discordantly pleiotropic, affecting both traits in opposite directions.

## Estimating polygenicity with MiXeR – Trivariate

There is a large degree of a trivariate genetic overlap across MDD, BPD, and ADHD (Figs 3–4). The trivariate mixture model estimates highlight ADHD as a potentiator of comorbidity, as consistent with the literature. In the three-way mixture

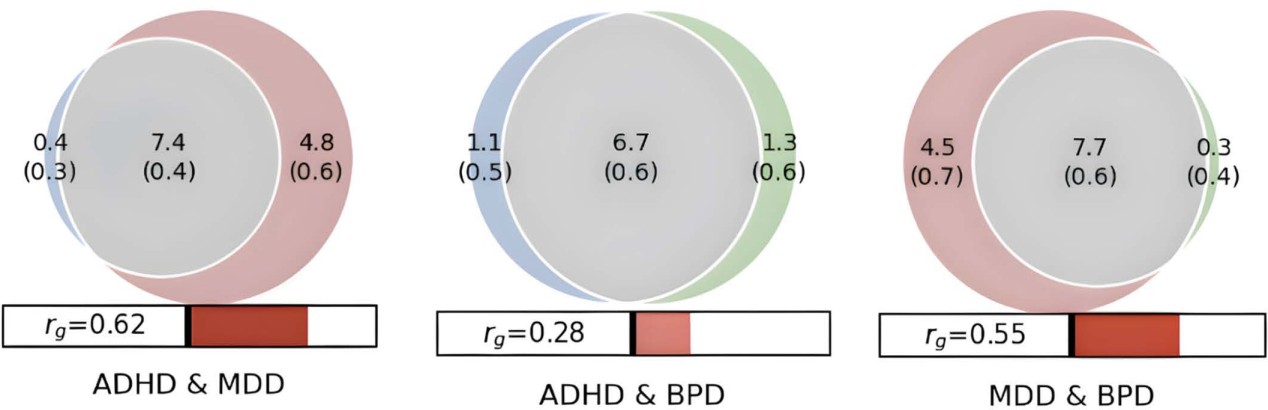

**Fig 2. Bivariate genetic overlap across ADHD (Blue), MDD (Red), and BPD (Green), with MiXeR estimates of genetic correlation.** Shared genetic variants are colored grey. Venn diagram numbers represent the amount of causal variants (In thousands), that explain 90% of SNP heritability with standard deviations in parenthesis.

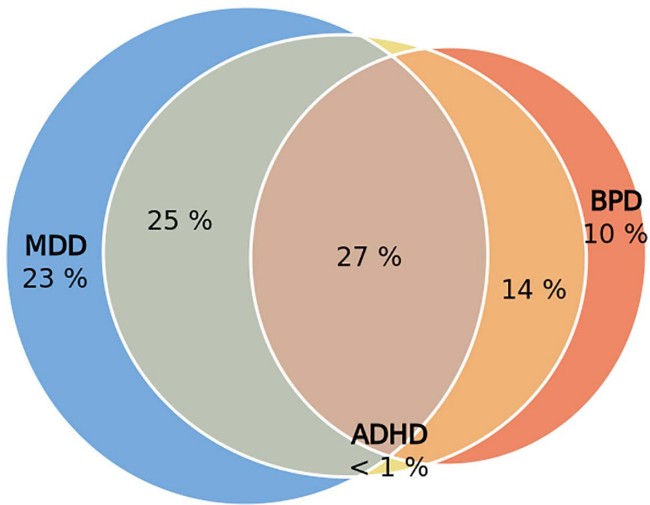

**Fig 3. Euler diagram from trivariate mixer analysis of ADHD, BPD, and MDD, denoting the percentage distribution of shared and trait-specific genetic causal variants across the 3 psychiatric disorders.**

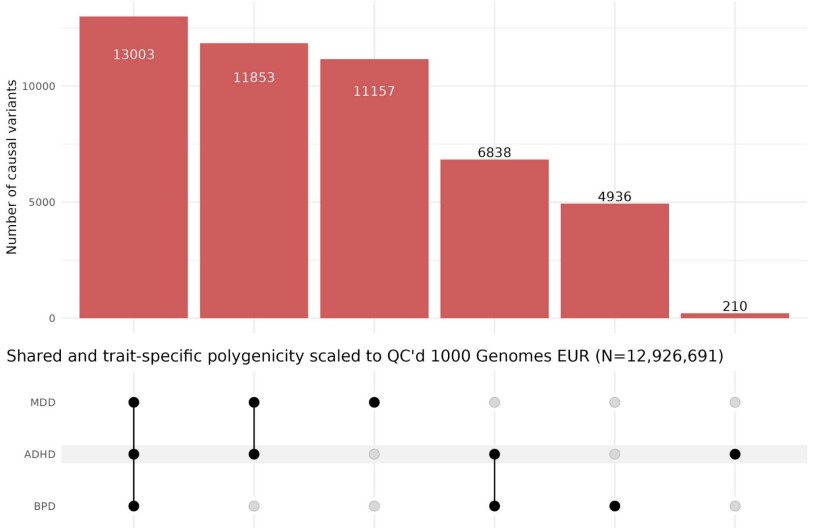

**Fig 4. UpSet plot of shared and trait-specific polygenicity across ADHD, MDD, and BPD, scaled to QC'd 1000 Genomes European reference panel (MAF > 0.001, INFO > 0.8, call rate > 0.9, and hardy weinberg equilibrium p-value > 5e-10, N = 12926691), from trivariate mixer causal variant proportions.**

model, less than 1% of the nc@9p variants that make up ADHD are intrinsically specific to ADHD. A majority of the causal variants that influence ADHD are not unique to the disorder itself, but are shared with MDD and BPD.

## Multivariate GWAS

We fit a latent factor model indexing the shared genetic liability across MDD, BPD, and ADHD (F1). The model was fully saturated (df = 0), an exact reparameterization of the empirical genetic covariance matrix into a structural equation model. F1 loaded significantly on all traits (Unstandardized loading on MDD = 1, ADHD = 1.07, BPD = 0.94). The loading of MDD was fixed to 1 for factor identification and scaling, allowing the variance of F1 to be freely estimated. There were a total of 16132 genome-wide SNP associations with F1 (P-value ≤ 5e-8), all of which represent a novel association for at least one of the constituent disorders. We characterized linkage-disequilibrium independent hits with a clumping threshold of $r^2 < 0.1$ in a 1Mb window. F1 was associated with 350 independent genomic risk loci, 105 of which were not previously identified by the contributing univariate GWASs (S31 Table). The mean $\chi^2$ and $\lambda_{GC}$ of the multivariate summary statistics were estimated at 2.27 and 1.87, respectively. However, an LDSC-intercept of 1.015 suggests that genomic control and mean $\chi^2$ inflation are a result of polygenic heritability as opposed to confounding factors such as population stratification or sample overlap (Figs 5 and 6).

## Characterization of novel associations identified by F1

Utilizing the midbrain dopaminergic Hi-C enriched H-MAGMA annotation, F1 was enriched in 1384 genes (Bonferroni correction, p-value threshold of 9.7e-07) (Fig 7, S3 Table). Notably, each of these genes represent a novel association for at least one of the constituent disorders. Across MDD, BPD, and ADHD, a total of 2936 gene associations identified in the F1 H-MAGMA analysis were undetected using univariate summary statistics. We performed over-representation analysis (Fisher's exact test) using WebGestalt [26] to characterize both novel and previously detectable associations for each disorder (Fig 8). The univariate summary statistics of ADHD and novel MDD gene associations were not sufficiently powered to carry out either ORA or GSEA, no gene-sets passed multiple testing correction (Bonferroni correction and FDR < 0.05,

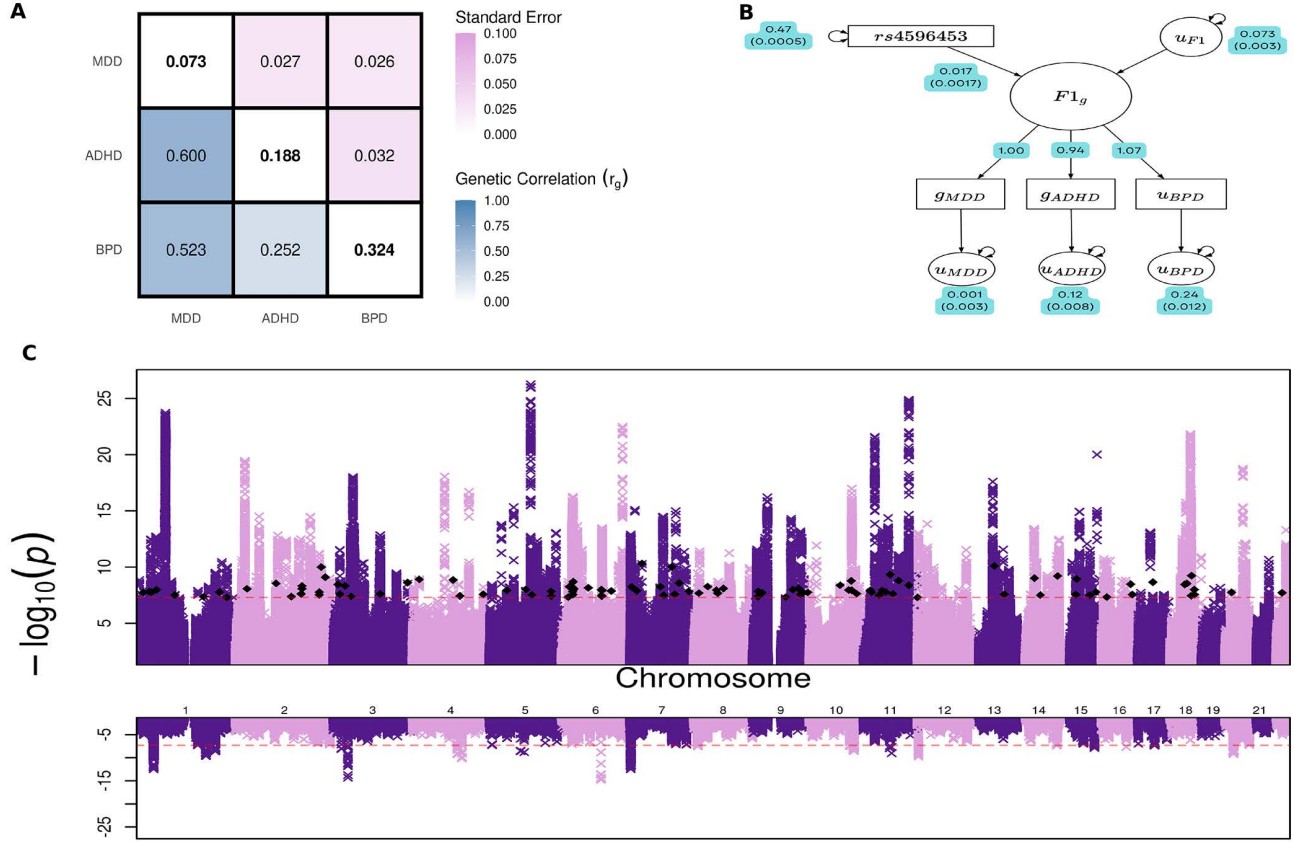

**Fig 5. Genomic SEM analysis of shared genetic architecture and locus discovery for the latent factor F1. (A)** Matrix of total liability scale bivariate genetic correlations ($r_g$) attained from LDSC in Genomic SEM, where diagonal elements are total liability scale SNP heritability ($h_2$), bottom off diagonal elements are total liability scale bivariate genetic correlations, and top off diagonal elements are bottom off diagonal standard errors. **(B)** Path diagram depicting how index SNP rs2585813 influences the latent factor F1 from genomic SEM analysis using WLS estimation. All path coefficients are standardized with standard errors in parentheses. In the diagram, "g" subscripts denote genetic variables and "u" subscripts denote the residual variance of their respective genetic variables not explained by F1. The effect of SNP rs2585813 on F1 was 0.017 (SD = 0.00017). **(C)** Miami plot of genome wide association results for F1 with p-value thresholds of 5e-08 for both panels, represented by the red dotted line. The top panel displays the -$log_{10}$(p-value) for each SNP with black diamonds denoting independent risk loci (P < 5e-8) that were not previously detectable by the contributing GWASs. The bottom panel displays the SNP's respective -$log_{10}$($Q_{SNP}$ p-value) from the heterogeneity test ($Q_{SNP}$ test estimates the degree to which a SNP is not mediated by F1)..

respectively). We report the top associations (Fig 8), however these findings should be interpreted with caution. After applying H-MAGMA to the univariate summary statistics, there were 107 associations for ADHD (S11 Table), 378 associations for BPD (S10 Table), and 1235 for MDD (S9 Table). Applying H-MAGMA to F1, we identified 1331 novel associations for ADHD, 1209 novel associations for BPD, and 396 novel associations for MDD. Significant gene-sets of novel and previously detectable associations are reported in S12–S17 Tables.

**Gene, gene-set, and celltyping analysis.** Using MAGMA's default annotation that maps SNPs to genes via predefined gene coordinates, including SNPs 35kb upstream and 10kb downstream [24], a total of 28 gene sets reached genome-wide significance (Table 2). A significant proportion of genome-wide significant gene-sets are involved in various dynamics of the neuronal synapse. In all brain regions and cell-types assayed, F1 was enriched to varying degrees. Most notably, F1 was significantly enriched in the cerebellum and in cholinergic neurons (Figs 9–10). In the multivariate transcriptome-wide analysis, F1 was enriched in a total of 272 genes across GTExv8 brain regions [31] (P-value < 1e-06) (Fig 11).

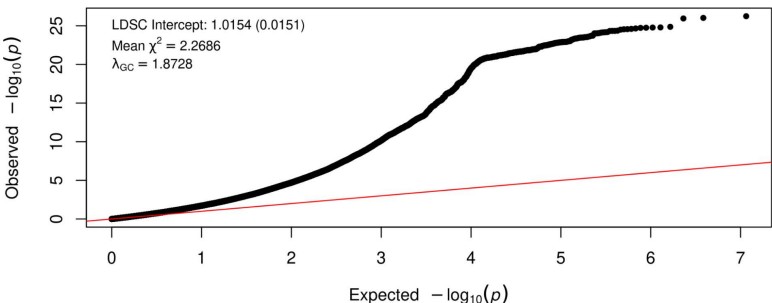

**Fig 6. QQ plot of expected vs. observed -log$_{10}$(p-value) for the F1 summary statistics with LDSC intercept, mean chi-squared, and lambda.**

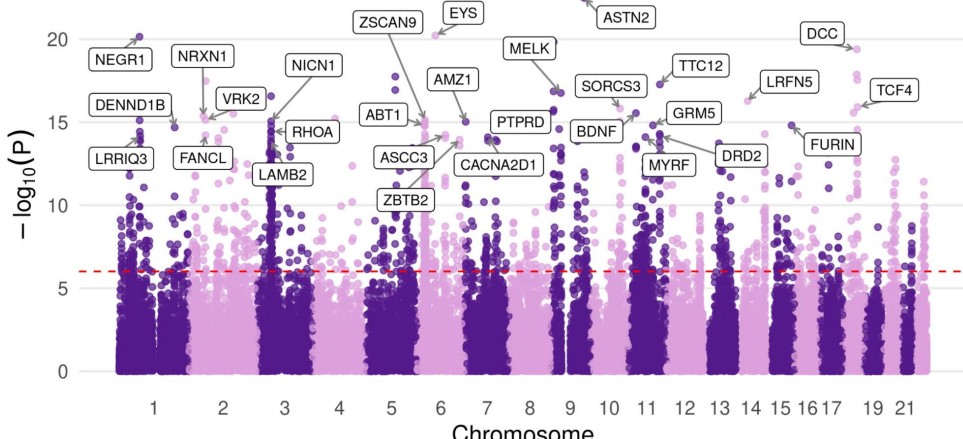

**Fig 7. Manhattan plot of F1 gene associations from DA neuron H-MAGMA analysis.** The top 30 protein coding genes are labeled, and the Bonferroni adjusted p-value threshold is represented by the dotted line (p = 9.7e-07).

## Candidate gene biological annotation – *PTPRD*

Protein tyrosine phosphatase receptor delta (*PTPRD)*, was among the top 10 protein coding genes associated with F1, p = 2e-16 (0.05/51684 genes tested = p-value threshold of 9.67e-07). *PTPRD* encodes a receptor tyrosine phosphatase with cell adhesion mediating immunoglobulin and fibronectin III domains and a catalytically active D1 phosphatase domain [32]. Given its statistical prioritization in the dopaminergic H-MAGMA analysis, we evaluated if recent literature was consistent with a potential role in neuronal pathways relevant to F1.

### Review of *PTPRD* KO mice studies and DA-related behavioral alteration

Among all studies that assay reward-related behaviors, *PTPRD* KO/Inhibited mice exhibited consistent deviation from the wild-type [32–35] (Table 3). We next considered how *PTPRD* knockout and inhibition significantly altered reward-related behaviors such as cocaine-conditioned place preference, cocaine self-administration, motivation for cocaine, and goal-oriented behavior particularly in the context of DA's involvement in reward processing and *PTPRD*'s association with F1 in the dopaminergic Hi-C enriched gene analysis (See Supplementary Information in S1 Appendix).

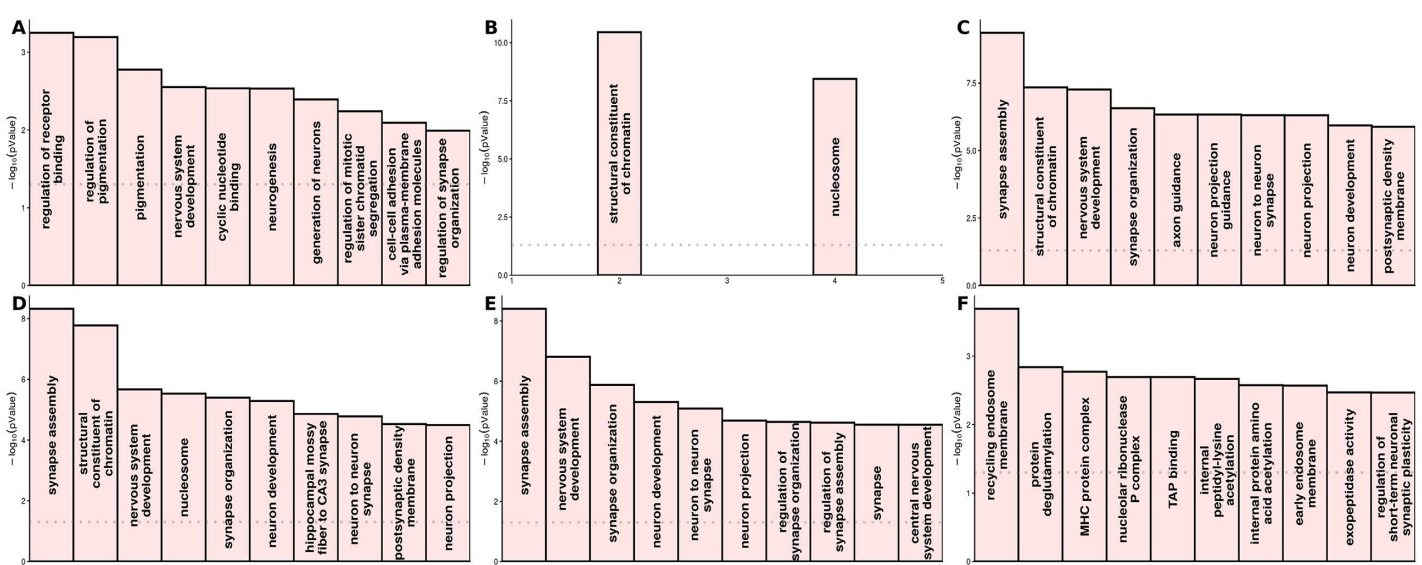

**Fig 8. Top 10 gene ontology terms for novel and previously detectable dopaminergic gene associations from the H-MAMGA analysis identified by Fisher's exact test using WebGestalt. (A)** Top gene ontology terms for previously detectable dopaminergic associations using ADHD summary statistics. **(B)** Top gene ontology terms for previously detectable dopaminergic associations using BPD summary statistics. **(C)** Top gene ontology terms for previously detectable dopaminergic associations using MDD summary statistics. **(D)** Top gene ontology terms for novel ADHD dopaminergic associations identified by F1. **(E)** Top gene ontology terms for novel BPD dopaminergic associations identified by F1. **(F)** Top gene ontology terms for novel MDD dopaminergic associations identified by F1.

## Discussion

We estimated a latent factor indexing the shared genetic liability of BPD, MDD, and ADHD, F1. At the genome-wide level, we observed significant genetic overlap across the disorders. We identified 350 independent genomic risk loci associated with F1, 105 of which were not previously reported by the contributing univariate GWASs. Pleiotropic variants were largely involved in various dynamics of the neuronal synapse, suggesting that neurotransmission heavily underlies the observed genetic covariances. Utilizing the Allen Human Brain Atlas RNA-expression dataset, F1 was significantly enriched in cholinergic neurons and in the cerebellum. We identified significant genetic overlap across the disorders, with ADHD and BPD exhibiting discordant pleiotropy amongst shared causal variants. Additionally, we performed an in-depth biological annotation of the candidate gene *PTPRD*, speculating a mechanistic cascade that characterizes its potential role in disease pathogenesis.

We reviewed existing literature to assess the plausibility of *PTPRD*'s role in dopaminergic neurotransmission. *PTPRD* may contribute to the chronic excitation of DA neurons through increased thresholds of structural long-term potentiation and inhibitory long-term depression, as well as decreased DAT-mediated synaptic dopamine uptake. These mechanisms may help explain the consistent trend of altered behavioral responses to cocaine in *PTPRD*-deficient mice (Supplementary Note 15 in S1 Appendix). Further research is required to validate our conclusions, therefore we remain cautious in our interpretations. Additionally, we observed a spike in LD spanning 10.9Mb to 12Mb on chromosome 9 (S13 Fig). Functionally relevant domains containing expression quantitative trait loci (eQTLs), methylation quantitative trait loci (meQTLs), and chromatin loops exhibit reduced recombination [36]. Given the concurrence of abnormally high SNP heritability and reduced recombination, this region may represent a DA neuron regulatory domain, housing cis-eQTLs for *PTPRD* and a plethora of other genes (Supplementary Note 16 in S1 Appendix).

In the context of multivariate GWAS-driven genetic discovery, there remains a substantial amount of undiscovered associations [37]. This is especially true for GWASs with small sample sizes, for which a great number of causal SNPs

**Table 2. Table of F1 genome-wide significant gene-sets using MAGMA's default gene annotation file.**

| GENE ONTOLOGY TERMS | NGENES | BETA | BETA_STD | SE | P |
|---|---|---|---|---|---|
| SYNAPTIC MEMBRANE | 423 | 0.37065 | 0.032653 | 0.056666 | 3.0993E-11 |
| CENTRAL NERVOUS SYSTEM NEURON DIFFERENTIATION | 236 | 0.46189 | 0.030447 | 0.074716 | 3.2015E-10 |
| SYNAPSE ASSEMBLY | 252 | 0.45618 | 0.031068 | 0.074169 | 3.9041E-10 |
| MECHANOSENSORY BEHAVIOR | 16 | 1.7804 | 0.030619 | 0.30177 | 1.8382E-09 |
| GABAERGIC SYNAPSE | 104 | 0.61696 | 0.02703 | 0.10573 | 2.7087E-09 |
| SYNAPSE | 1599 | 0.17149 | 0.029049 | 0.030421 | 8.7174E-09 |
| PRESYNAPTIC MEMBRANE | 170 | 0.49894 | 0.027931 | 0.08871 | 9.3812E-09 |
| POSTSYNAPTIC MEMBRANE | 299 | 0.37074 | 0.027491 | 0.066994 | 1.5773E-08 |
| SYNAPSE ORGANIZATION | 540 | 0.27206 | 0.02705 | 0.05018 | 2.9728E-08 |
| STRIATUM DEVELOPMENT | 20 | 1.5352 | 0.029518 | 0.28951 | 5.7399E-08 |
| POSTSYNAPTIC DENSITY MEMBRANE | 117 | 0.55478 | 0.025777 | 0.10783 | 1.346E-07 |
| LEARNING | 157 | 0.47546 | 0.025581 | 0.092627 | 1.4334E-07 |
| DENDRITIC TREE | 591 | 0.24322 | 0.025287 | 0.047546 | 1.573E-07 |
| POSTSYNAPTIC SPECIALIZATION MEMBRANE | 144 | 0.49196 | 0.025353 | 0.09657 | 1.759E-07 |
| CELL JUNCTION ASSEMBLY | 488 | 0.27162 | 0.025685 | 0.053545 | 1.9724E-07 |
| NEURON TO NEURON SYNAPSE | 382 | 0.29063 | 0.02434 | 0.058101 | 2.85E-07 |
| SOMATODENDRITIC COMPARTMENT | 806 | 0.20292 | 0.024588 | 0.040727 | 3.1553E-07 |
| CELL JUNCTION ORGANIZATION | 801 | 0.20528 | 0.024797 | 0.041417 | 3.6095E-07 |
| GENERATION OF NEURONS | 1482 | 0.15277 | 0.024941 | 0.030935 | 3.9575E-07 |
| SUBPALLIUM DEVELOPMENT | 26 | 1.1325 | 0.024827 | 0.2304 | 4.451E-07 |
| NEUROGENESIS | 1738 | 0.13894 | 0.024505 | 0.028649 | 6.2093E-07 |
| SYNAPTIC SIGNALING | 798 | 0.19925 | 0.024025 | 0.041422 | 7.5654E-07 |
| POSTSYNAPTIC SPECIALIZATION | 367 | 0.28848 | 0.023684 | 0.060162 | 8.171E-07 |
| POSTSYNAPSE | 737 | 0.20323 | 0.023563 | 0.043125 | 1.228E-06 |
| NEURON PROJECTION | 1276 | 0.15242 | 0.023136 | 0.032888 | 1.7948E-06 |
| REGULATION OF SYNAPSE STRUCTURE OR ACTIVITY | 298 | 0.31093 | 0.023017 | 0.067436 | 2.0139E-06 |
| ABNORMAL_MORPHOLOGY OF THE LIMBIC SYSTEM | 49 | 0.86265 | 0.025955 | 0.18819 | 2.2891E-06 |
| BEHAVIOR | 671 | 0.2045 | 0.022638 | 0.044965 | 2.7164E-06 |

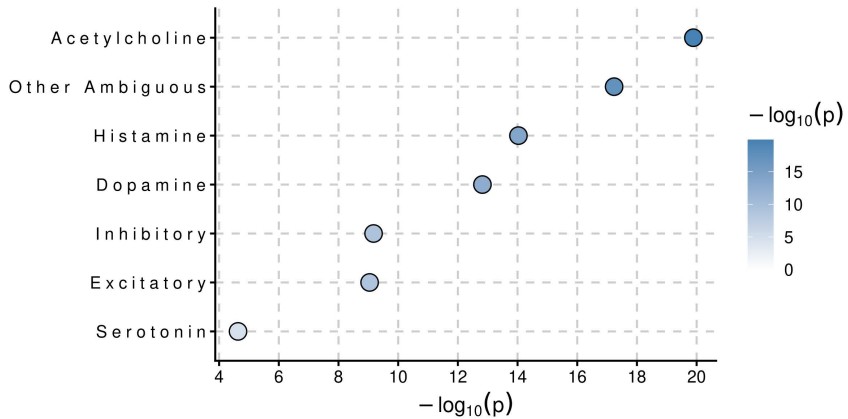

**Fig 9. Dotplot of clustered cell types associated with F1 from MAGMA Cell Typing analysis using the Human Brain Atlas v1.0 RNA expression dataset.**

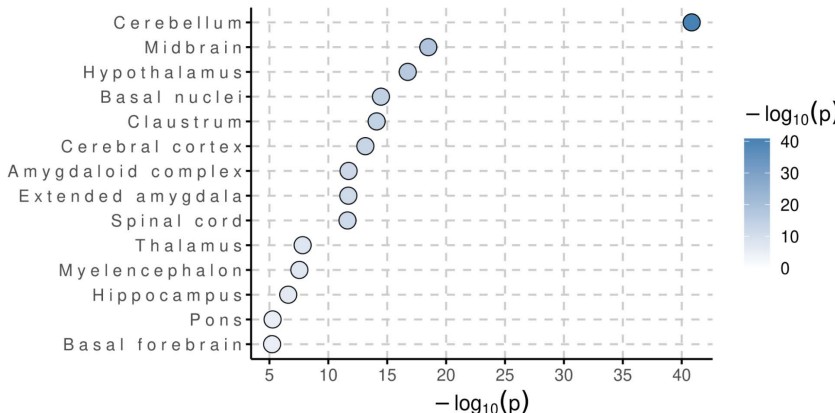

**Fig 10. Dotplot of brain regions associated with F1 from MAGMA Cell Typing analysis using the Human Brain Atlas v1.0 RNA expression dataset.**

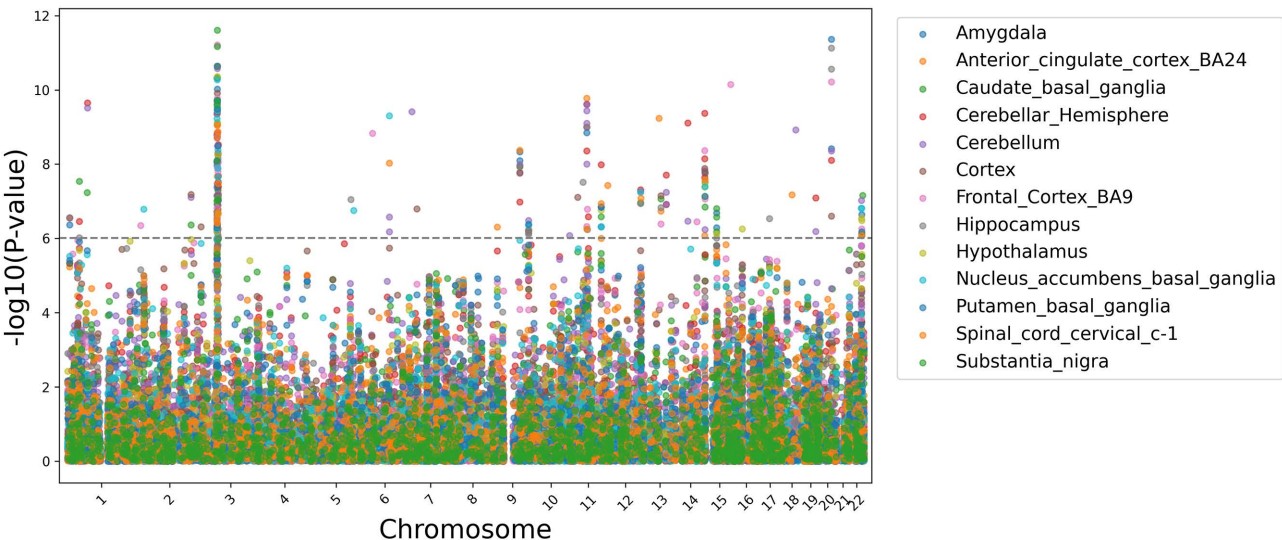

**Fig 11. Manhattan plot of F1 TWAS gene associations from GTExv8 brain regions.** The significance threshold is denoted by the grey dotted line.

**Table 3. Reward-associated behavioral outcomes of *PTPRD* KO and inhibition in animal studies in comparison to the WT throughout existing literature.**

| Study | Phenotype | Behavioral Impact |
|---|---|---|
| Uhl et al. 2018 | *PTPRD* heterozygous KO mice; acute *PTPRD* inhibition (7-BIA) in mice | Reduced cocaine reward in self administration and conditioned place-preference |
| Drgonova et al. 2015 | *PTPRD* heterozygous KO mice | Reduced cocaine-conditioned place preference at high dosage and increased at cocaine-conditioned place preference at low dosage |
| Ho et al. 2023 | *PTPRD* heterozygous and homozygous KO mice | Reduced goal oriented behavior (Nest building) |
| Uhl & Martinez, 2019 | Review of human genetics, mouse KO, and pharmacological studies of *PTPRD* | Reduced *PTPRD* expression associated with decreased stimulant reward |

do not reach genome-wide significance due to lack of power. It is important to note that multivariate approaches cannot identify trait-specific associations, nor associations that lack substantial effect size from at least one of the contributing GWASs. Dopaminergic genes that have been implicated with experimental support but lacked SNP heritability across all disorders like *SELENOT* in ADHD [38], *DRD3* in BPD [39], and the DAT transporter in MDD [40] did not perform well in the latent factor model. In contrast, well characterized dopaminergic genes that were significantly enriched for at least one of the constituent disorders such as *DCC*, *BDNF*, *TCF4*, and *TTC12* [41–44], performed extremely well. Multivariate GWASs leverage the genetic covariance structure between trait(s) to detect pleiotropic loci. Over 80% of the case sample size of F1 originated from the MDD summary statistics. Leveraging the strong genetic covariances with MDD substantially increased power and identified a combined 2540 novel dopaminergic gene associations for BPD and ADHD that were not detectable with the respective univariate summary statistics. Across all three disorders, a total of 2936 novel associations were identified. We observed distinct biological mechanisms between novel associations identified by F1 and previously detectable associations using the univariate summary statistics. F1 significantly informs ADHD and BPD through pleiotropic associations involved in synapse assembly and organization, nervous system development, and neuron to neuron synapses. These gene-sets sharply contrasted to ADHD and BPD-specific mechanisms characterized by regulation of receptor binding and chromatin constituent functionality, respectively.

There are several limitations to our study. First, SEM's do not reveal direct associations for the contributing GWASs. Rather, latent factors represent the underlying genetic covariance structure. Second, the statistical methods employed in this study carry out association tests and cannot be interpreted with causality. For all candidates identified, establishing direct causal relationships would require extensive wet lab validation experiments. Third, our analysis was restricted to European ancestry due to the limited availability of well-powered GWAS summary statistics for other groups. As a result, the generalizability of our findings to other ancestries is limited. The genetic architecture of BPD, ADHD, and MDD may differ across different ancestral groups due to discrepancies in linkage disequilibrium structure, allele frequency, and population stratification. Concurrently, most analytical frameworks for GWAS data require genetically homogenous cohorts.

## Supporting information

**S1 Fig. RNA expression tables displaying mean UMIs for *NTRK2* in 10 subclasses of human dopamine neuron clusters from the Kamath et al. 2022 dataset.**
(TIFF)

**S2 Fig. RNA expression tables displaying mean UMIs for *PTPRD* in 10 subclasses of human dopamine neuron clusters from the Kamath et al. 2022 dataset.**
(TIFF)

**S3 Fig. RNA expression tables displaying mean UMIs for *PDGFRB* in 10 subclasses of human dopamine neuron clusters from the Kamath et al. 2022 dataset.**
(TIFF)

**S4 Fig. RNA expression tables displaying mean UMIs for *KCNQ2* in 10 subclasses of human dopamine neuron clusters from the Kamath et al. 2022 dataset.**
(TIFF)

**S5 Fig. RNA expression tables displaying mean UMIs for *VAV2* in 10 subclasses of human dopamine neuron clusters from the Kamath et al. 2022 dataset.**
(TIFF)

**S6 Fig. RNA expression tables displaying mean UMIs for *RAC1* in 10 subclasses of human dopamine neuron clusters from the Kamath et al. 2022 dataset.**
(TIFF)

**S7 Fig. RNA expression tables displaying mean UMIs for *CAMK2A* in 10 subclasses of human dopamine neuron clusters from the Kamath et al. 2022 dataset.**
(TIFF)

**S8 Fig. RNA expression tables displaying mean UMIs for *CAMK2B* in 10 subclasses of human dopamine neuron clusters from the Kamath et al. 2022 dataset.**
(TIFF)

**S9 Fig. RNA expression tables displaying mean UMIs for *CAMK2D* in 10 subclasses of human dopamine neuron clusters from the Kamath et al. 2022 dataset.**
(TIFF)

**S10 Fig. RNA expression tables displaying mean UMIs for *CAMK2G* in 10 subclasses of human dopamine neuron clusters from the Kamath et al. 2022 dataset.**
(TIFF)

**S11 Fig. RNA expression tables displaying mean UMIs for *RET* in 10 subclasses of human dopamine neuron clusters from the Kamath et al. 2022 dataset.**
(TIFF)

**S12 Fig. RNA expression tables displaying mean UMIs for *PRKCB* in 10 subclasses of human dopamine neuron clusters from the Kamath et al. 2022 dataset.**
(TIFF)

**S13 Fig. Linkage disequilibrium heatmap of *PTPRD* extended (Chr 9: 7.4Mb to 12Mb), consisting of the top off-diagonals of an $r^2$ matrix calculated using the 1000 Genomes Project EUR reference panel.**
(TIFF)

**S14 Fig. F1 Manhattan plot of *PTPRD* extended with the *PTPRD* risk locus bounds denoted by the red dotted lines and standardized hg19 PTPRD coordinates denoted by the green dotted lines.**
(TIFF)

**S15 Fig. Linkage disequilibrium heatmap of *PTPRD* extended with the *PTPRD* risk locus highlighted by the green dotted line (Chr 9, 10968693–11709873), consisting of the top off-diagonals of an $r^2$ matrix calculated using the 1000 Genomes Project EUR reference panel.**
(TIFF)

**S16 Fig. Linkage disequilibrium heatmap of PTPRD risk locus, consisting of the top off-diagonals of an $r^2$ matrix calculated using the 1000 Genomes Project EUR reference panel.**
(TIFF)

**S1 Table. MAGMA gene-level association results for the latent factor F1 using the adult brain Hi-C annotation.**
(XLSX)

**S2 Table. MAGMA gene-level association results for the latent factor F1 using the adult cortical Hi-C annotation.**
(XLSX)

**S3 Table. MAGMA gene-level association results for the latent factor F1 using the adult midbrain Hi-C annotation.**
(XLSX)

**S4 Table. MAGMA gene-level association results for the latent factor F1 using the fetal brain Hi-C annotation.**
(XLSX)

**S5 Table. MAGMA gene-level association results for the latent factor F1 using the iPSC-derived neuronal Hi-C annotation.**
(XLSX)

**S6 Table. MAGMA gene-level association results for the latent factor F1 using the default gene model.**
(XLSX)

**S7 Table. MAGMA gene-set enrichment analysis results for the latent factor F1.**
(XLSX)

**S8 Table. MAGMA Q_SNP heterogeneity statistics identifying loci with disorder-specific effects deviating from the latent factor F1.**
(XLSX)

**S9 Table. MAGMA gene-level association results for major depressive disorder using adult midbrain Hi-C annotation.**
(XLSX)

**S10 Table. MAGMA gene-level association results for bipolar disorder using adult midbrain Hi-C annotation.**
(XLSX)

**S11 Table. MAGMA gene-level association results for attention-deficit/hyperactivity disorder using adult midbrain Hi-C annotation.**
(XLSX)

**S12 Table. WebGestalt pathway enrichment results for dopaminergic genes associated with major depressive disorder.**
(XLSX)

**S13 Table. WebGestalt pathway enrichment results for novel dopaminergic gene associations identified by F1 for major depressive disorder.**
(XLSX)

**S14 Table. WebGestalt pathway enrichment results for dopaminergic genes associated with attention-deficit/hyperactivity disorder.**
(XLSX)

**S15 Table. WebGestalt pathway enrichment results for novel dopaminergic gene associations identified by F1 for attention-deficit/hyperactivity disorder.**
(XLSX)

**S16 Table. WebGestalt pathway enrichment results for dopaminergic genes associated with bipolar disorder.**
(XLSX)

**S17 Table. WebGestalt pathway enrichment results for novel dopaminergic gene associations identified by F1 for bipolar disorder.**
(XLSX)

**S18 Table. Transcriptome-wide association study results for the latent factor F1 using the GTEx v8 European amygdala dataset.**
(XLSX)

**S19 Table. Transcriptome-wide association study results for the latent factor F1 using the GTEx v8 European anterior cingulate cortex (BA24) dataset.**
(XLSX)

**S20 Table. Transcriptome-wide association study results for the latent factor F1 using the GTEx v8 European caudate nucleus dataset.**
(XLSX)

**S21 Table. Transcriptome-wide association study results for the latent factor F1 using the GTEx v8 European cerebellar hemisphere dataset.**
(XLSX)

**S22 Table. Transcriptome-wide association study results for the latent factor F1 using the GTEx v8 European cerebellum dataset.**
(XLSX)

**S23 Table. Transcriptome-wide association study results for the latent factor F1 using the GTEx v8 European cortical dataset.**
(XLSX)

**S24 Table. Transcriptome-wide association study results for the latent factor F1 using the GTEx v8 European frontal cortex (BA9) dataset.**
(XLSX)

**S25 Table. Transcriptome-wide association study results for the latent factor F1 using the GTEx v8 European hippocampal dataset.**
(XLSX)

**S26 Table. Transcriptome-wide association study results for the latent factor F1 using the GTEx v8 European hypothalamic dataset.**
(XLSX)

**S27 Table. Transcriptome-wide association study results for the latent factor F1 using the GTEx v8 European nucleus accumbens dataset.**
(XLSX)

**S28 Table. Transcriptome-wide association study results for the latent factor F1 using the GTEx v8 European putamen dataset.**
(XLSX)

**S29 Table. Transcriptome-wide association study results for the latent factor F1 using the GTEx v8 European cervical spinal cord dataset.**
(XLSX)

**S30 Table. Transcriptome-wide association study results for the latent factor F1 using the GTEx v8 European substantia nigra dataset.**
(XLSX)

**S31 Table. LD-independent genome-wide significant SNPs (P<5e-8) associated with the latent factor F1 after clumping at r²<0.1.** The column univ_GWAS_sig denotes which phenotypes the respective SNP association reached GWAS significance for. (XLSX)

**S32 Table. Fine-mapped cis-credible variants for F1 at the cis-PTPRD locus identified by MiXeR-finemap.** (XLSX)

**S33 Table. Cell-type enrichment analysis of latent factor F1 using the Adult Human Brain Cell Atlas curated by the Allen Institute.** (XLSX)

**S34 Table. Brain-region enrichment analysis of genes associated with the latent factor F1 using the Adult Human Brain Cell Atlas curated by the Allen Institute.** (XLSX)

**S1 Appendix. Supplementary Information.** (DOCX)

## Acknowledgments

Anna K. Lack[1]

## Author contributions

**Conceptualization:** Christopher Lawrence, Thomas Folkmann Hansen.

**Data curation:** Christopher Lawrence.

**Formal analysis:** Christopher Lawrence.

**Investigation:** Christopher Lawrence.

**Methodology:** Christopher Lawrence, Thomas Folkmann Hansen.

**Project administration:** Christopher Lawrence.

**Resources:** Christopher Lawrence.

**Software:** Christopher Lawrence, Thomas Folkmann Hansen.

**Supervision:** Christopher Lawrence, Thomas Folkmann Hansen.

**Validation:** Christopher Lawrence.

**Visualization:** Christopher Lawrence.

**Writing – original draft:** Christopher Lawrence.

**Writing – review & editing:** Christopher Lawrence.

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
