## [Decision Letter · Decision Letter 0]

30 Nov 2025

Dear Dr. Lawrence,

Thank you for submitting your manuscript to PLOS ONE. After careful consideration, we feel that it has merit but does not fully meet PLOS ONE’s publication criteria as it currently stands. Therefore, we invite you to submit a revised version of the manuscript that addresses the points raised during the review process.

We look forward to receiving your revised manuscript.

Kind regards,

Stephen D. Ginsberg, Ph.D.

Section Editor

PLOS ONE

Journal Requirements:

2. Please upload a new copy of Figure 16 as the detail is not clear. Please follow the link for more information:  https://journals.plos.org/plosone/s/figures

3. We notice that your supplementary figures are included in the manuscript file. Please remove them and upload them with the file type 'Supporting Information'. Please ensure that each Supporting Information file has a legend listed in the manuscript after the references list.

Reviewers' comments:

Reviewer's Responses to Questions

**Comments to the Author**

1. Is the manuscript technically sound, and do the data support the conclusions?

Reviewer #1: Partly

Reviewer #2: Partly

2. Has the statistical analysis been performed appropriately and rigorously?

Reviewer #1: Yes

Reviewer #2: Yes

3. Have the authors made all data underlying the findings in their manuscript fully available?

Reviewer #1: Yes

Reviewer #2: Yes

4. Is the manuscript presented in an intelligible fashion and written in standard English?

Reviewer #1: No

Reviewer #2: Yes

Reviewer #1: The manuscript by Lawrence and Hansen utilized open databases to explore shared genetic liability across MDD, ADHD, and BPD. My comments are as follows:

1) The references should be grouped, e.g. (A) (B) (C) to (A; B; C). Using software like Endnote or Reference Manager would be helpful. Additionally, the title is too long.

2) Do not prepare the figures in small, individual pieces. Please refer to a paper published in Cell or Science, and use it as an example to rationally organize your information into larger figures. The same goes for your tables. Currently, they look like screenshots from your Excel drafts rather than properly formatted tables for publication.

3) The current format is not standard for a scientific paper. The Results section does not even mention PTPRD, yet the Discussion is suddenly all about it. Please move and organize all your findings, noted with (Supplementary) Figure X and Table X, into the Results section rather than in the Discussion section.

4) In the Discussion section, please discuss your overall findings, implications, methods, and limitations by comparing them to other studies’ findings. Include genes previously demonstrated in dopamine neurons of humans or transgenic animals associated with these diseases, such as dopaminergic SELENOT in ADHD (https://pubmed.ncbi.nlm.nih.gov/40195499/), and discuss how these genes performed in your model. I would also prefer seeing a discussion regarding the phenotypes of animals with PTPRD (conditional) knockout or overexpression.

5) The Abstract: Your hypotheses and proposals may be suitable for the Discussion section. However, the majority of the Abstract should focus on your findings rather than your thoughts. Again, please find an example to guide you in write the manuscript.

Reviewer #2: SUMMARY

This manuscript by Lawrence and Hansen presents a comprehensive multivariate genome-wide association study investigating the shared genetic architecture of major depressive disorder (MDD), bipolar disorder (BPD), and attention-deficit hyperactivity disorder (ADHD). Using publicly available summary statistics from the Psychiatric Genomics Consortium, the authors employ multiple statistical frameworks including MiXeR, Genomic SEM, H-MAGMA, and MAGMA Cell Typing to identify pleiotropic genetic variants and genes. The study identifies substantial genetic overlap across the three disorders, with a latent factor explaining shared liability. Notably, the multivariate analysis reveals 1,025 novel gene associations in dopaminergic neurons that were undetected in univariate analyses, with PTPRD emerging as a top candidate gene. The authors propose a mechanistic model whereby PTPRD loss-of-function leads to dysregulated dopaminergic neurotransmission through hyperactivation of TrkB and RET signaling pathways, ultimately affecting reward processing and contributing to psychiatric comorbidity. While the methodological approach is rigorous and the genetic discovery findings are valuable, the manuscript requires significant improvements in figure presentation, more cautious interpretation of mechanistic claims, and expanded discussion of the multivariate findings.

MAJOR COMMENTS

1. The figures require substantial revision to meet publication standards:

• Figures 2-3 (MAF comparison plots): Correlation values should be directly annotated on each panel rather than only in titles. Consider adding regression lines and confidence intervals.

• Figure 4 (Venn diagram): The visual proportions appear inconsistent with the stated genetic correlations. For example, ADHD-BPD show 80%+ overlap but rg=0.28. Please clarify this apparent discrepancy or revise the visualization.

• Figures 9 and 11 (Manhattan plots): These lack clear visual distinction between novel discoveries and previously reported genes. Consider using different colors or symbols, and highlighting the top novel genes.

• Figure 10 (QQ plot): The substantial deviation from the null (λGC = 1.87) requires explicit discussion in Results and/or Methods.

• Figures 16-17 (Cell type enrichment): These would benefit from hierarchical clustering and clearer grouping of related cell types/brain regions.

• All figures need larger, more readable font sizes and colorblind-friendly palettes.

2. PTPRD Mechanistic Insights: The Discussion section presents an extensive mechanistic model for PTPRD function, but the strength of claims exceeds the evidence:

• The H-MAGMA analysis identifies PTPRD association in dopaminergic neurons, but this is based on Hi-C data and does not prove cell-type-specific function in humans.

• The mechanistic cascade (TrkB/RET → ERK1/2/VAV2/PKC → DAT → anhedonia) is highly speculative and based largely on knockout mouse studies and in vitro experiments.

• Lines 358-373 use deterministic language ("results in," "leads to") when the evidence supports only possibilities.

3. The identification of 1,025 novel gene associations (Figure 15) is perhaps the most important finding, yet it receives minimal discussion:

• No analysis of why these genes were undetected in univariate analyses (power vs. opposite-direction effects)

• No examination of whether novel genes have distinct biological functions compared to known genes

• Missing discussion of which specific disorder(s) each novel gene association informs

• Add a dedicated Results subsection characterizing the 1,025 novel genes

• Perform pathway enrichment analysis comparing novel vs. previously detected genes

• Show effect direction concordance across MDD/BPD/ADHD for top novel hits

• Discuss implications for understanding shared vs. distinct mechanisms

**Do you want your identity to be public for this peer review?** For information about this choice, including consent withdrawal, please see our Privacy Policy

Reviewer #1: No

Reviewer #2: No

---

## [Author Response · Author response to Decision Letter 1]

18 Jan 2026

We thank the Editor and Reviewers for their constructive comments. We have carefully addressed all points raised in the decision letter and reviewer reports. A detailed, point-by-point response is provided in the accompanying Response to Reviewers document, and all revisions have been incorporated into the manuscript.

Reviewer #1: The manuscript by Lawrence and Hansen utilized open databases to explore shared genetic liability across MDD, ADHD, and BPD. My comments are as follows:

1) The references should be grouped, e.g. (A) (B) (C) to (A; B; C). Using software like Endnote or Reference Manager would be helpful. Additionally, the title is too long.

We have revised the references and adopted grouping suggestions. Additionally, we have changed the title so that it is significantly shorter and better reflects the objective of the manuscript. The manuscript title is now “Dissecting the shared genetic architecture of bipolar disorder, major depressive disorder, and attention-deficit hyperactivity disorder”, changed from “Protein tyrosine phosphatase receptor type D - A convergent risk gene for MDD, BPD, and ADHD and regulator of dopaminergic neuroplasticity in reward associated circuits; A multivariate GWAS, brain tissue and cell type enrichment, and gene fine-mapping.”

2) Do not prepare the figures in small, individual pieces. Please refer to a paper published in Cell or Science, and use it as an example to rationally organize your information into larger figures. The same goes for your tables. Currently, they look like screenshots from your Excel drafts rather than properly formatted tables for publication.

We have consolidated and made multi-panel figures. More specifically, we have combined figures 1 and 2 in addition to combining figures 7-9. We have removed all Excel screenshots and replaced them with tables properly formatted for publication.

3) The current format is not standard for a scientific paper. The Results section does not even mention PTPRD, yet the Discussion is suddenly all about it. Please move and organize all your findings, noted with (Supplementary) Figure X and Table X, into the Results section rather than in the Discussion section.

Thank you for this clarification. We have organized all findings, most importantly moving the biological annotation of PTPRD noted with (Supplementary) Figure X and Table X to the results section.

4) In the Discussion section, please discuss your overall findings, implications, methods, and limitations by comparing them to other studies’ findings. Include genes previously demonstrated in dopamine neurons of humans or transgenic animals associated with these diseases, such as dopaminergic SELENOT in ADHD (https://pubmed.ncbi.nlm.nih.gov/40195499/), and discuss how these genes performed in your model. I would also prefer seeing a discussion regarding the phenotypes of animals with PTPRD (conditional) knockout or overexpression.

Thank you for noticing these inconsistencies. We have included dopaminergic genes with existing literature that support its association with the constituent phenotype(s). See lines 300-305. Regarding the phenotypes of animals with PTPRD KO or inhibition, we have implemented a table into the PTPRD biological annotation within the results section. This table characterizes the phenotypes and behavioral outcomes from existing literature.

5) The Abstract: Your hypotheses and proposals may be suitable for the Discussion section. However, the majority of the Abstract should focus on your findings rather than your thoughts. Again, please find an example to guide you in write the manuscript.

Thank you for the suggestion. In the revised abstract, we have focused on the most important findings of the manuscript. These topics include independent loci identified by F1, trait-specific and shared polygenicity, novel dopaminergic gene associations for the contributing disorders, and cell type and brain tissue enrichment. Most importantly, we have refrained from a long discussion on PTPRD and omitted general thoughts.

MAJOR COMMENTS

1. The figures require substantial revision to meet publication standards:

• Figures 2-3 (MAF comparison plots): Correlation values should be directly annotated on each panel rather than only in titles. Consider adding regression lines and confidence intervals.

Thank you for this suggestion, we have overlaid the respective correlation values onto each panel rather than in the title. We have added the regression confidence interval to both figures (See figure 1 which consists of the revised multi-panel).

• Figure 4 (Venn diagram): The visual proportions appear inconsistent with the stated genetic correlations. For example, ADHD-BPD show 80%+ overlap but rg=0.28. Please clarify this apparent discrepancy or revise the visualization.

Thank you for addressing this obscurity. We have added some clarification in the results section (See lines 148-151). The large discrepancy between causal variant overlap and genetic correlation is suggestive of discordant pleiotropy. This is typical when traits share a large proportion of causal variants but with opposite effect directions.

• Figures 9 and 11 (Manhattan plots): These lack clear visual distinction between novel discoveries and previously reported genes. Consider using different colors or symbols, and highlighting the top novel genes.

We added a visual component to figure 9 (Which is now a component of a larger figure, figure 5C), including novel independent loci (SNPs with p-value < 5e-8 and r2 > 0.2) identified by F1. These loci did not reach genome-wide significance in any of the contributing GWASs and are clearly denoted by black diamonds.

• Figure 10 (QQ plot): The substantial deviation from the null (λGC = 1.87) requires explicit discussion in Results and/or Methods.

Thank you for pointing out this discrepancy. We have added explicit discussion in the results (See lines 182-184). The lambda was 1.87, however an intercept of 1.015 suggests that the inflation of lambda and mean chi squared was due to polygenicity as opposed to confounding factors like sample overlap or population stratification.

• Figures 16-17 (Cell type enrichment): These would benefit from hierarchical clustering and clearer grouping of related cell types/brain regions.

Thank you for this suggestion. We have clustered cells by neurotransmitter system, significantly improving the clarity and interpretability of figure 16.

• All figures need larger, more readable font sizes and colorblind-friendly palettes.

Thank you for this suggestion. We have increased the font sizes for figures 1, 2, 7, 8, and 13 (Figures 1 and 2 have been consolidated into figure 1 A&B, figures 7 and 8 have been consolidated into figure 5 A&B, respectively, and figure 13 is now figure 9).

2. PTPRD Mechanistic Insights: The Discussion section presents an extensive mechanistic model for PTPRD function, but the strength of claims exceeds the evidence:

• The H-MAGMA analysis identifies PTPRD association in dopaminergic neurons, but this is based on Hi-C data and does not prove cell-type-specific function in humans.

• The mechanistic cascade (TrkB/RET → ERK1/2/VAV2/PKC → DAT → anhedonia) is highly speculative and based largely on knockout mouse studies and in vitro experiments.

• Lines 358-373 use deterministic language ("results in," "leads to") when the evidence supports only possibilities.

Thank you for addressing this issue. Indeed, the evidence only supports possibilities. We have significantly reduced causal language to keep the interpretations cautious and within the bounds of our findings.

3. The identification of 1,025 novel gene associations (Figure 15) is perhaps the most important finding, yet it receives minimal discussion:

• No analysis of why these genes were undetected in univariate analyses (power vs. opposite-direction effects)

Thank you for this suggestion. MAGMA does not consider the direction of SNP effects, it is simply an aggregation. These genes went undetected in the univariate analyses solely because the GWASs were underpowered.

• No examination of whether novel genes have distinct biological functions compared to known genes

Thank you for this suggestion, we have examined distinct biological functions identified by Fisher’s exact test (A hypergeometric test, elucidating if a list of candidate genes appear in curated gene-sets more than what would be expected by chance). See lines 312-316.

• Missing discussion of which specific disorder(s) each novel gene association informs

• Add a dedicated Results subsection characterizing the 1,025 novel genes

• Perform pathway enrichment analysis comparing novel vs. previously detected genes

Thank you for these suggestions, we have implemented a new results section including Fisher’s exact test using WebGestalt to characterize novel and previously detectable gene associations for each of the individual disorders. We have also added a discussion (See lines 204-219).

• Show effect direction concordance across MDD/BPD/ADHD for top novel hits

Thank you for this suggestion. MAGMA does not consider the directional effects of genetic variants, it simply aggregates them. All Z scores are positive.

• Discuss implications for understanding shared vs. distinct mechanisms

We have addressed the implications for understanding shared vs. distinct mechanisms as it relates to the intrinsic limitations of multivariate approaches. Multivariate approaches leverage genetic covariances to increase power to identify pleiotropic loci. Even in the case of well characterized dopaminergic genes with support of existing literature, if these do not reach genome-wide significance in any of the univariate GWASs (Due to lack of power), they do not perform well in multivariate analyses. In contrast, genes that both reach genome-wide significance for at least one trait and have effect concordance typically perform very well.

---

## [Decision Letter · Decision Letter 1]

29 Jan 2026

Dissecting the shared genetic architecture of bipolar disorder, major depressive disorder, and attention-deficit hyperactivity disorder

PONE-D-25-50065R1

Dear Dr. Lawrence,

We’re pleased to inform you that your manuscript has been judged scientifically suitable for publication and will be formally accepted for publication once it meets all outstanding technical requirements.

Kind regards,

Stephen D. Ginsberg, Ph.D.

Section Editor

PLOS One

**Comments to the Author**

Reviewer #1: All comments have been addressed

Reviewer #2: All comments have been addressed

2. Is the manuscript technically sound, and do the data support the conclusions?

Reviewer #1: Yes

Reviewer #2: Yes

3. Has the statistical analysis been performed appropriately and rigorously?

Reviewer #1: I Don't Know

Reviewer #2: Yes

4. Have the authors made all data underlying the findings in their manuscript fully available?

Reviewer #1: Yes

Reviewer #2: Yes

5. Is the manuscript presented in an intelligible fashion and written in standard English?

Reviewer #1: Yes

Reviewer #2: Yes

Reviewer #1: I have no further comments regarding the manuscript Dissecting the shared genetic architecture of bipolar disorder, major depressive disorder, and attention-deficit hyperactivity disorder.

Reviewer #2: Thank you for the responses to all of my previous comments. All questions have been fully addressed. I noticed one minor formatting issue in Figure 5 that would benefit from correction prior to final acceptance. Specifically, in Figure 5A and Figure 5C, some of the axis/label text appears stretched or distorted, which slightly affects readability. Could you please adjust the figure formatting so that the labels render cleanly and proportionally?

**Do you want your identity to be public for this peer review?**  For information about this choice, including consent withdrawal, please see our Privacy Policy

Reviewer #1: No

Reviewer #2: No

---

## [Editor Report · Acceptance letter]

PONE-D-25-50065R1

PLOS One

Dear Dr. Lawrence,

I'm pleased to inform you that your manuscript has been deemed suitable for publication in PLOS One. Congratulations! Your manuscript is now being handed over to our production team.

Kind regards,

on behalf of

Dr. Stephen D. Ginsberg

Section Editor

PLOS One